# Under My Skin: Reducing Bias in STEM through New Approaches to Assessment of Spatial Abilities Considering the Role of Emotional Regulation

**Michelle Lennon-Maslin \*, Claudia Michaela Quaiser-Pohl** 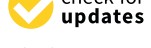**, Vera Ruthsatz and Mirko Saunders**

Department of Developmental Psychology and Psychological Assessment, Faculty of Educational Sciences
Institute of Psychology, University of Koblenz, 56070 Koblenz, Germany; quaiser@uni-koblenz.de (C.M.Q.-P.);
ruthsatz@uni-koblenz.de (V.R.); mirkosaunders@uni-koblenz.de (M.S.)
**\*** Correspondence: mlennonm@uni-koblenz.de

**Abstract:** Reducing gender bias in STEM is key to generating more equality and contributing to a more balanced workforce in this field. Spatial ability and its components are cognitive processes crucial to success in STEM education and careers. Significant gender differences have consistently been found in mental rotation (MR), the ability to mentally transform two- and three-dimensional objects. The aim of this pilot study is to examine factors in psychological assessment which may contribute to gender differences in MR performance. Moreover, findings will inform the development of the new approaches to assessment using computer adaptive testing (CAT). (1) Background: The study examines the impact of emotional regulation on MR performance in primary school children whose mean age was 9.28 years old. (2) Methods: Skin conductance was measured to assess the impact of emotional reactivity (ER) on performance during an MR task. (3) Results: Patterns of ER influence response time (RT) on specific items in the task. (4) Conclusions: Identifying the effects of emotional arousal and issues of test construction such as stereotyped stimuli and item difficulty in tests of spatial ability warrants ongoing investigation. It is vital to ensure that these factors do not compromise the accurate measurement of performance and inadvertently contribute to the gender gap in STEM.

**Keywords:** gender stereotypes; gender gap; STEM; spatial abilities; emotional regulation; skin conductance; CAT



## 1. Introduction

Despite globalization, gendered career choices persist worldwide (Makarova et al. 2019). The World Economic Forum (2019) estimates that currently only one third of female students pursue higher education or research careers in the fields of science, technology, engineering and mathematics (STEM) (World Economic Forum 2019).

The current study, being undertaken at the University of Koblenz, Germany, is funded by the EU-financed (Horizon 2020) research network SellSTEM (Spatially Enhanced Learning linked to STEM). Spatial ability and its components are cognitive processes crucial to success in the STEM arena (Newcombe 2017). The SellSTEM network, consisting of 15 PhD candidates and their supervisors from 10 universities across Europe as well as 8 partners, is investigating the role of spatial ability in STEM learning.

The rationale for this research is based on three key findings: (1) too few young people are enrolling in STEM courses (Maselli and Beblavý 2014); (2) in scientific fields such as psychology, life and social sciences, women are present in much higher numbers; and (3) there isa widely-agreed-upon consensus that in undergraduate and postgraduate university programs, which are the most mathematically intensive, such as geoscience, engineering, economics, mathematics/computer science and the physical sciences, women are still underrepresented (Ceci et al. 2014). Furthermore, the percentage of women in some

STEM faculties, for example biomedical engineering, at the assistant, associate and full professor levels, remains low (Chesler et al. 2010).

Although gender differences on spatial tests on a broader scale have been found to be small or less consistent, there are subsets or components of spatial ability which continue to yield significant differences in favor of males (Voyer et al. 1995; Linn and Petersen 1985).

The aim of the four-year project is to raise spatial ability in children, and girls in particular, in a number of European countries, that is, Latvia, Norway, Ireland, Germany, The Netherlands, Sweden, Austria and the UK, so that they are better prepared for the cognitive demands of STEM education. Ultimately, the goal of the working group is to promote more successful STEM learning, triggering migration in larger numbers toward STEM careers and consequently generating a more gender-balanced ratio in the field (SellSTEM MSCA ITN 2021). A specific goal of the SellSTEM subproject based at the University of Koblenz is to provide primary school teachers across Europe with effective ways of assessing spatial ability development in children with the objective that such methods will assist them in identifying potential difficulties in STEM learning, thus facilitating early intervention.

### 1.1. Mental Rotation Tests (MRTs): Assessing a Spatial Ability

Mental rotation (MR), a component of spatial ability, has been studied extensively in psychology and education due to significant gender differences found in the ability to rotate mental representations of two- and three-dimensional objects (Neuburger et al. 2012b). Despite a small amount of evidence on cognitive disparity in spatial ability as a whole, MR tasks, particularly those incorporating three-dimensional objects, consistently yield large and reliable differences in performance favoring males, with no significant reduction (Wraga et al. 2006).

MR is traditionally assessed using psychometric instruments (often referred to as paper-and-pencil tests) such as the mental rotation test (VMRT; Vandenberg and Kuse 1978). In a bid to modernize psychometric assessment, many of these tests have become digitalized (Schmand 2019). During a computerized MRT, participants respond to each item or series of stimuli on a computer or touch-screen device (Monahan et al. 2008). Stimuli usually take the form of pictorial representations of abstract objects, such as cubical or pellet-shaped figures, and concrete objects such as animals or toys (Rahe and Quaiser-Pohl 2021). The stimuli are rotated in-depth or in picture-plane.

Recently, researchers examining MR testing have turned their attention to issues which may influence participants' performance. Due to the aforementioned gender differences, a number of studies have focused on how MR test characteristics, item and stimulus attributes might contribute to this phenomenon, e.g., time limitations (Rahe and Quaiser-Pohl 2021), item difficulty and rotational axis (Neuburger et al. 2012b), task-solving strategies (Saunders and Quaiser-Pohl 2020) and gender-stereotyped stimuli (Ruthsatz et al. 2015).

Performance on tests of spatial ability and other cognitive tasks is known to be influenced by the test-taker's experience and their emotional state (Shepard and Feng 1972; Johns et al. 2008). Moreover, findings demonstrate that emotion-regulating processes reduce working memory, and executive resources needed to perform well on tests of cognitive ability (Schmader et al. 2008). For instance, Ramirez and colleagues (2012) found that some younger children reported experiencing a phenomenon referred to as spatial anxiety (SA), that is, feeling anxious at the prospect of engaging in spatial activities. Furthermore, pronounced gender differences in SA and self-confidence have been found to mediate gender differences in MR performance, especially when task demands are high (Arrighi and Hausmann 2022).

Therefore, this pilot study is a first step in the development of new approaches to MR assessment in children. These approaches may reduce the negative effects of emotions such as anxiety and have a positive influence on self-confidence in participants. The study thereby examines factors beyond mental rotation itself, such as emotional regulation, stereotype threat and task and item difficulty, which are known to contribute to gender

differences in performance on an MRT (Sanchis-Segura et al. 2018; Fladung and Kiefer 2016; Caissie et al. 2009).

### 1.2. Computer Adaptive Tests of MR (CAT-MR): An Alternative to Fixed Item Tests (FITs)

Computer adaptive testing (CAT) is an approach to psychometrics, which establishes a link between the participant's ability, their response to items and the underlying trait being measured (Veldkamp and Verschoor 2019). During CAT, a computer algorithm automatically presents items to participants and selects the next item based on their previous response. As opposed to fixed item testing (FIT), in which participants respond to the same set of items in the same order (Vispoel 1993), CAT adapts to their performance (Linden et al. 2000). Moreover, CAT has been found to elicit less participant anxiety (Fritts and Marszalek 2010; Ortner and Caspers 2011) and female primary school students achieved better results, reported a higher sense of motivation and a more positive subjective test experience after CAT (Martin and Lazendic 2018). In mathematics, CAT revealed promising results regarding the reduction in stress and anxiety, and a better overall performance among test candidates (Eggen and Verschoor 2006).

Moreover, findings demonstrate that CAT offers teachers many advantages such as convenience and flexibility, faster, more accurate scoring and reporting, potentially shorter tests, reduced scheduling and supervision, fewer test items needed to accurately estimate proficiency and less time needed for marking (Chuesathuchon 2008).

### 1.3. The Effects of Stereotype Threat on Cognitive Performance

More than a decade of research has now confirmed that experiencing negative stereotypes about one's social group can impair an individual's ability to achieve their potential (Schmader 2010). This effect is referred to as stereotype threat (ST) or situations in which individuals perceive themselves to be at risk of conforming to negative stereotypes about their ingroup (Steele and Aronson 1995). The beneficial effect of perceiving oneself as conforming to positive stereotypes about one's ingroup is known as stereotype lift (SL) (Walton and Cohen 2003).

ST has been found to arouse anxiety in targets, and in the process of emotional regulation, cognitive resources needed to successfully execute a task are thereby limited (Schmader et al. 2008). Consequently, performance on MR tests is susceptible to the effects of ST (Neuburger et al. 2012a). SL, on the other hand, may alleviate self-doubt, anxiety and fear of rejection that could otherwise hinder performance on MR and other cognitive tests (Walton and Cohen 2003).

Gender-stereotyped beliefs influence boys' and girls' preferences and behaviors early in life. This can often determine their choice of clothes and hairstyles and can also influence their preference for certain toys (Ruthsatz et al. 2019). Boys' preference for construction toys and other games that are related to object manipulation in space might lead to more practice in these skills than playing with stereotypically female toys, such as, dolls and cuddly toys (Newcombe and Frick 2010). This increased familiarity with construction-related play might be advantageous to male participants on an MR task consisting of items with the frequently used cubical figure (Kersh et al. 2008).

More recently, some studies have focused on the gender-stereotyped attributes of the stimuli used in MR tests and how they might impact a test-taker's performance (Rahe and Quaiser-Pohl 2019). Findings suggest that the degree to which stimulus objects are familiar to male versus female participants, that is, are gender-congruent, is a significant determinant of the gender difference (Neuburger et al. 2011). Moreover, a positive correlation between stereotyped stimulus content and children's performance on tests of MR has been established (Ruthsatz et al. 2014).

### 1.4. Assessing the Impact of Emotional Regulation on MR in Children

Performance on MR and other cognitive tasks is known to be heavily influenced by the participant's experience (Shepard and Feng 1972) and emotional state (Schmader et al.

2008). Various studies have examined the source of emotional arousal experienced by participants undertaking cognitive tests such as an MRT. As previously mentioned, the effects of stereotype threat can negatively impact cognitive performance (Johns et al. 2008), as can the mode of testing, e.g., FIT or CAT (Vispoel 1993). Additionally, anticipatory feelings, such as spatial anxiety, regarding the kinds of activities a test may involve are important factors which mediate performance (Ramirez et al. 2012).

Human cognitive functions such as perception, attention, memory and problem-solving skills are all significantly influenced by emotion (Tyng et al. 2017). Events or stimuli which evoke emotion are classified according to two main categories, valence and arousal (Costanzi et al. 2019). On a continuum ranging from negative to neutral to positive, the attractiveness or aversiveness of a stimulus or event is referred to as valence. An event or stimulus intensity is known as arousal when it ranges from being intensely calming to intensely exhilarating or agitating (Costanzi et al. 2019). Moreover, how an event or stimulus is evaluated plays a role in judgment and decision-making. Positive emotion surrounding an event or stimulus enhances thinking and reasoning and adversely negative emotion impairs these functions (Storbeck and Clore 2008). The relevance and salience of a stimulus or event to the observer also determines how it is prioritized. Emotional arousal enhances cognition for high-priority information and impairs it for that of low-priority (Turkileri et al. 2021).

Tests and examinations, for example, are often perceived as events which can elicit a variety of emotional states, including excitement, frustration, anxiety and even boredom (Tyng et al. 2017). As is evident from emotional experiences such as spatial anxiety, the emotional reaction to the subject matter itself can impair an individual's test performance (Tyng et al. 2017). Similarly, the stimuli which make up a test item can impact valence and arousal in ways that can either enhance or impair cognitive performance on that test (Costanzi et al. 2019). If these stimuli are not relevant to the observer, they will be assigned low priority; hence, cognition may be impaired by a lack of emotional arousal. Furthermore, individual differences in emotional regulation strategies have also been found to influence test performance. For example, in one study, the strategies used by males and females to regulate emotions during an MR test resulted in gender differences in their performance therein (Fladung and Kiefer 2016).

Investigating emotional experience arising in the course of an MR task could also be measured physiologically. Assessing somatic activity such as skin conductance and heart rate could provide valuable information with regard to differences in emotion expressive behavior, valence and arousal impacting performance on such tasks (Deng et al. 2016). Measurements of children's electrodermal and cardiovascular activity can be used as indicators of the autonomic nervous system's (ANS) activation during emotion-evoking events or when experiencing stimuli which may impact participants emotionally (Sohn et al. 2001).

Skin conductance responses (SCRs) are biomarkers of ANS arousal and are a well-established method for measuring psychophysiological functioning in humans (Christopoulos et al. 2019). These signals, stemming from the peripheral nervous system (PNS), have been long identified as important for mental functions, in particular emotions (James 1890). Furthermore, these somatic responses are likely an essential part of the emotional experience and act as cues based on which they are formed (Damasio 2001).

Galvanic skin response (GSR), a measure of skin conductance, has been successfully employed to measure physiological changes as a result of emotional expression such as anxiety and stress in children (Najafpour et al. 2017). Furthermore, the use of wearable devices to measure GSR can facilitate the analysis of physiological responses in children engaged in cognitive tasks in real-life conditions such as in the classroom (Geršak et al. 2019).

*1.5. The Current Pilot Study*

Using an experimental design, the goal of this study is to examine the impact of emotional regulation on spatial performance in primary school children during a computerized

MR test. It is anticipated that findings will also provide useful information about item characteristics and precedes the development of a computer adaptive test of mental rotation (CAT-MR).

### 1.5.1. Research Questions

Based on findings from previous research, this study aims to answer the following questions:

1. Will there be changes in participants' skin conductance in the course of a mental rotation task and will this be impacted by the difficulty of the task?
2. Will skin conductance levels influence participants' accuracy and reaction time on the MR task? Will participants who take more time demonstrate more accuracy on the MR task?
3. Will there be patterns in participants' speed and accuracy which relate to the item difficulty, i.e., stimulus type and rotational axis on a MR task?
4. Regarding responses on a stereotype nature of stimuli and a perceived difficulty questionnaire, will participants identify stereotyped and difficult stimuli and will their responses correspond with their performance on the task?

### 1.5.2. Hypotheses

**Hypotheses 1.** *Previous research has found that emotion regulation is an important factor for maintaining MR performance (Fladung and Kiefer 2016). Therefore, we hypothesize that there will be an increase in participants' skin conductance levels (SCL) measured by GSR during the MR task from baseline to subtest MRT1 and subtest MRT2.*

**Hypotheses 2.** *Item difficulty on an MR task is determined by stimulus type, that is, concrete or abstract objects, and rotational axis, that is, stimuli rotated in-depth or in picture-plane. Items containing abstract objects rotated in-depth are known to be more difficult in an MR task (Neuburger et al. 2012b). Furthermore, failure to effectively regulate negative emotional states, elicited, for example, by phenomena such as stereotype threat or spatial anxiety, can lead to a poorer performance on difficult cognitive tasks (Schmader 2010; Ramirez et al. 2012; Fladung and Kiefer 2016). Moreover, participants who spend more time and put more effort into solving difficult items on an MR test usually demonstrate better performance (Liesefeld et al. 2015). Therefore, we hypothesize that increased SCL measured by GSR and a longer reaction time (RT) predict higher scores on more difficult items, i.e., abstract stimuli rotated in-depth, on the MR task.*

**Hypotheses 3.** *Items with abstract objects rotated in-depth and gender-stereotyped objects can influence participants' accuracy and reaction time on an MR task. Mean accuracy scores are often higher on items with gender-congruent stimuli and lower on items with abstract stimuli rotated in-depth (Ruthsatz et al. 2019). Due to the difficulty level, reaction time is often longer on items with abstract stimuli rotated in-depth (Liesefeld et al. 2015). Furthermore, interaction between stimuli and emotional arousal and valence may account for some of these differences (Costanzi et al. 2019). Therefore, we hypothesize that there will be differences in accuracy scores and reaction time between two SCL groups on items with concrete and abstract stimuli rotated in picture-plane and in-depth and on three gendered objects, that is, male-stereotyped, female-stereotyped and gender-neutral objects, during the MR task.*

**Hypotheses 4.** *Long before an awareness of their own gender identity commences, children have already developed a schema of gender-associated traits and gender categories (Martin et al. 2002). Therefore, we hypothesize that participants will identify stimuli containing gender-stereotyped objects as such, that is, male-stereotyped objects as masculine and female-stereotyped objects as feminine.*

**Hypotheses 5.** *In line with previous findings, participants will identify stimuli containing concrete objects rotated in picture-plane as easier and abstract objects rotated in-depth as more difficult (Ruthsatz et al. 2017).*

## 2. Materials and Methods

### 2.1. Participants

For the pilot study, we recruited 29 students from third- and fourth-grade classes at two local primary schools in Koblenz, Rheinland Palatine, Germany. One student withdrew consent immediately prior to testing and three of the task data collected could not be retrieved from the devices. Therefore, the total number of participants whose data were used in the final analysis was 25 ($N = 25$). There were 12 children who identified as boys and 12 as girls and 1 participant who did not provide a response to the gender question. The average age of the students was 9.28 years old ($M = 9.28$). All 29 parents and guardians provided written informed consent.

### 2.2. Material and Instrumentation

#### 2.2.1. Mental Rotation Task

A computerized MR task (Vandenberg and Kuse 1978) was programmed in PsychoPy® software (Supplementary Materials) and installed on Microsoft Pro 8 Surface tablets, each with a keyboard and a mouse. The task was programmed to record both the number of correct responses (accuracy) as well as the time taken to answer each item (reaction time).

Items included MR stimuli for younger children, i.e., animals, letters and cubes (Quaiser-Pohl 2003) as well as abstract and concrete stimuli rotated in-depth or in picture-plane. Abstract items consisted of stimuli such as cubes, pellets (Ruthsatz et al. 2014) and polyhedra (Ruthsatz et al., forthcoming). Concrete items consisted of male and female gender-stereotyped stimuli (Ruthsatz et al. 2015) and gender-neutral stimuli (Ruthsatz et al., forthcoming). Examples of some of the items used are shown in Figure 1.

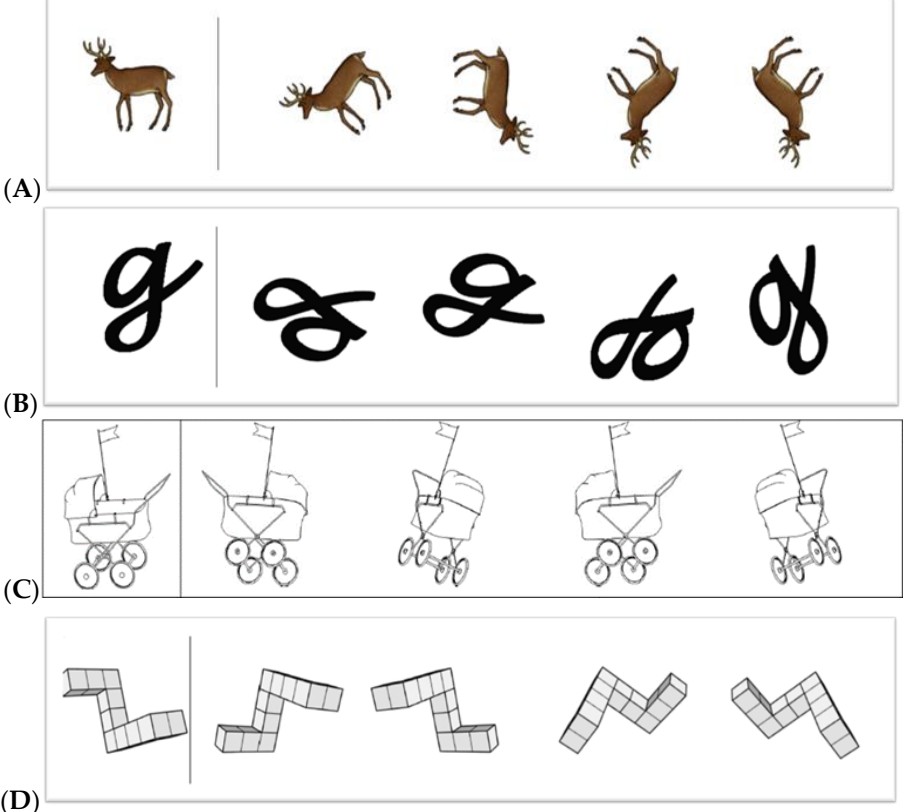

**Figure 1.** Examples of MRT items used with concrete objects: (**A**) animal stimulus and (**B**) letter stimulus (Hinze and Quaiser-Pohl 2003); (**C**) female-stereotyped stimulus "Pram" (Jansen et al. 2014; Ruthsatz et al. 2015) and with an abstract object (**D**) "Cube" figure (Ruthsatz et al. 2014).

The task was divided into two parts, each with a time limit: part one (MRT 1), considered an easier task due to stimuli rotated in picture-plane only, was limited to 5 min and part two (MRT 2), considered more difficult as items consisted of stimuli rotated in-depth, allowed participants 8 min to complete. MRT 1 had 6 abstract and 10 concrete items and MRT 2 had 6 abstract and 6 concrete items (See Appendix C). MRT 2 was also considered more difficult due to stimuli features, their complexity as well as rotational axis (Neuburger et al. 2015). Items were presented randomly in each part of the task with one target stimulus on the left and four comparison stimuli on the right. Participants were instructed to identify two out of four stimuli on the right which, although rotated, were identical to that on the left.

### 2.2.2. Gender-Stereotype Nature and Perceived Difficulty of Stimuli Questionnaires

Two self-reported questionnaires were also created in PsychoPy® and presented at the end of the MRT to assess the gender-stereotyped nature of stimuli and the stimuli-perceived difficulty.

Both scales are adapted from the Stereotyped Nature of Stimuli questionnaire (Neuburger et al. 2015). The second questionnaire uses emojis on a sliding 5-point scale in order to assess perceived difficulty of stimuli. The first point on the scale (1) represents easy or happy face emoji, and the fifth (5) difficult or sad face emoji. The third point (3) represents neither easy nor difficult or neutral face emoji.

The first point (1) on the stereotype questionnaire represents the rating more for boys, and the fifth (5) more for girls. The third point (3) is a gender-neutral rating. Both scales are appended to this report.

### 2.2.3. Demographic Data

An online questionnaire was presented to each participant at the beginning of the experiment to collect data relating to participants' age and gender.

### 2.2.4. Skin Conductance

Shimmer3 GSR+ Unit® was used to measure galvanic skin response (GSR). These devices were synchronized with ConsensysBasic® multi-sensor management software where they were calibrated for recording and a sampling rate of 5 Hz was set. A baseline recording of 2 min per participant was planned in order to compare this with GSR during the MR task. Only 24 GSR datasets in total were analyzed as 1 dataset was missing from the Shimmer device. Furthermore, in three datasets at baseline was not recorded on the device; therefore, they could not be included in the analysis of SCL across the three conditions—baseline, MRT 1 and MRT 2.

For the purposes of analyses, two SCL-level groups were created—low and high GSR—during the MR task. SCL levels vary individually in humans; therefore, following checks for normality and removal of outliers on the SCL variable, descriptive statistics were calculated in SPSS as follows: minimum SCL during the MR task was 0.01 μS per minute, maximum was 12.64 μS and mean SCL was 4.69 μS per minute ($M = 4.69$, $SD = 2.99$). Therefore, all values < 4.69 μS were categorized as low SCL ($N = 10$) and all values > 4.69 μS per minute were categorized as high SCL ($N = 11$) during the MR task. Skin conductivity is measured in units referred to as microsiemens (μS). "Micro" is a prefix meaning millionths, so 1 microsiemen (1 μS) is a unit of time in the International System of Units (SI) equal to one millionth of a siemen (Braithwaite et al. 2013).

### 2.3. Procedure

Approval for the pilot study was provided by the Ethics Committee of the University of Koblenz and also by the state authorities in Rheinland Palatine overseeing schools. Informed consent was sought and provided by parents and guardians of all students involved in the study. The class teacher and the principal also permitted the study to be conducted in the school.

The students were tested by two female researchers in a separate classroom with access to their teacher, if required. The room had adequate lighting and individual seating arrangements.

The researchers explained the MR task to the students by rotating objects such as a pair of scissors, a toy and a wooden object, while explaining that by turning this object around, it does not change its features. Students were then asked to imagine the object in their mind and when they could see it, try to rotate the object mentally. The purpose of the study and the significance of mental rotation in everyday life and for school work was also explained to the students. The researchers also checked in advance that students were familiar and comfortable with the use of a keyboard and a mouse. Furthermore, any student who required eye glasses was reminded to wear these while viewing the tablet screen.

Electrodes for measurement of GRS were attached to the fore- and index finder of the non-dominant hand of each participant. Shimmer devices were then switched on and synchronized with Consensys® software. Devices remained in their respective docks until the experiment could be initialized. Participants were advised to try not to move the hand to which the electrodes were attached.

Shimmer devices were then removed from each dock and a two-minute baseline measurement was recorded, after which the computerized experiment was initialized. Students were given time to read the on-screen instructions with the help of the researchers. After task-understanding was confirmed, an initial practice run followed, which contained three sample items providing feedback on accuracy.

When the practice run was complete, students were presented with further on-screen instructions, informing them that the first part of the test was about to begin and that there would be a 5 min time limit on this part of the MR task. They were also advised to try to work as fast and as accurately possible. In addition, students in each of the small groups were asked to wait until their peers could see a red stop sign which would appear at the end of MRT 1 before proceeding to MRT 2. The researchers kept track of the time throughout the experiment. The same procedure was repeated for the second part of the task.

At the end of the experiment, Shimmer devices were returned to their respective docks to end recording and electrodes were removed from participants' fingers. Data were labelled and imported to Consensys, then exported and saved on each respective table.

### 2.4. Data Analysis

Quantitative data analyses were performed on SPSS® 29 software for statistical analysis. A repeated-measures ANOVA was run to establish whether skin conductance level (SCL) differed significantly between the three test conditions—pre-test baseline, picture-plane and in-depth conditions. All participants took part in the three test conditions. The data in each of the groups were normally distributed. Sphericity was not violated and the Greenhouse–Geisser correction was reported as it is more stringent.

A Pearson coefficient correlation was used to determine whether there was a relationship between skin conductance, reaction time and scores on more difficult items, that is, items containing abstract objects rotated in-depth. Arising from this, multiple regression was run, entering those variables that produced a significant result to examine the way in which these variables relate to each other. The assumptions for using regression were checked and confirmed, i.e., the criterion variable was always continuous; the Mahalanobis distance values indicated that there were no substantial outliers; the residual scores were normally distributed and not related to the predicted values; and tolerance values did not exceed 0.2, indicating that there was no multi-collinearity.

One-way repeated-measures ANOVA were carried out to investigate the differences in skin conductance conditions across the MR task. Furthermore, accuracy and reaction time on items containing concrete and abstract stimuli rotated in-depth and in picture-plane and items containing gendered stimuli across the two SCl levels were analyzed. Observations were independent of one another and the sample was completely random. The independent variables were categorical on three and two levels and the dependent

variables were continuous and scale variables. Significant outliers were removed from the dependent variables. Where sphericity was violated, the Greenhouse–Geisser correction was applied (Tabachnick and Fidell 2013).

## 3. Results

### 3.1. Skin Conductance Levels across the MR Task (H1)

A repeated-measures ANOVA using the Greenhouse–Geisser correction showed that skin conductance level (SCL) differed significantly between the three test conditions—pre-test baseline, MRT 1 (only rotations in picture-plane) and MRT 2 (only in-depth rotations) ($F(2, 40) = 9.46$, $p = 0.004$)—with a small to medium effect size ($\eta^2 = 0.241$). As a result, 24% of variation in skin conductance level can be explained by the different test conditions. See Table 1.

**Table 1.** Tests of within-subjects effects of skin conductance conditions (SCL levels) from pre-test baseline SCL to MRT1 and MRT2 on the MR task.

| | Source | Type III Sum of Squares | df | Mean Square | F | Sig. | Partial Eta Squared |
|---|---|---|---|---|---|---|---|
| | | **Measure: SCL_Levels** | | | | | |
| factor1 | Sphericity Assumed | 17.104 | 2 | 8.552 | 6.348 | 0.004 | 0.241 |
| | Greenhouse–Geisser | 17.104 | 1.752 | 9.764 | 6.348 | 0.006 | 0.241 |
| error (factor1) | Sphericity Assumed | 53.888 | 40 | 1.347 | | | |
| | Greenhouse–Geisser | 53.888 | 35.034 | 1.538 | | | |

Specifically, pairwise-comparisons-highlighted SCL during the MRT 2 condition was significantly higher than in the baseline condition ($M = 5.31$, $p = 0.018$, CI (95%) 0.182–2.233). Moreover, SCL in the MRT 1 condition was significantly higher than in the baseline condition ($M = 5.10$, $p = 0.009$, CI (95%) 0.233–1.701). There was no significant difference between SCL in the MRT 1 and SCL in the MRT 2 condition. Therefore, it can be concluded that SCL increases during an MR test. However, task difficulty did not have a statistically significant impact on SCL in this sample. See Table 2.

**Table 2.** Pairwise comparisons of skin conductance conditions (SCL levels) from pre-test baseline SCL to MRT1 and MRT2 on the MR task.

| (I) Factor1 | (J) Factor1 | Mean Difference (I-J) | Std. Error | Sig. [a] | Lower Bound | Upper Bound |
|---|---|---|---|---|---|---|
| | | | | **Measure: SCL_Levels** | 95% Confidence Interval for Difference [a] | |
| 1 | 2 | −0.962 * | 0.283 | 0.009 | −1.701 | −0.223 |
| | 3 | −1.207 * | 0.393 | 0.018 | −2.233 | −0.182 |

* The mean difference is significant at the 0.05 level. [a]. Adjustment for multiple comparisons: Bonferroni.

Figure 2 graphically illustrates the mean SCL across three conditions from pre-test baseline SCL to SCL in MRT 1 and MRT 2. The error bars represent the standard error (SE), which indicates the variability of the estimated means within each condition. Those error bars representing MRT 1 and MRT 2 overlap, which suggests that the observed differences between the conditions are not statistically significant. This conclusion is supported by the results of the statistical analysis. See Figure 2.

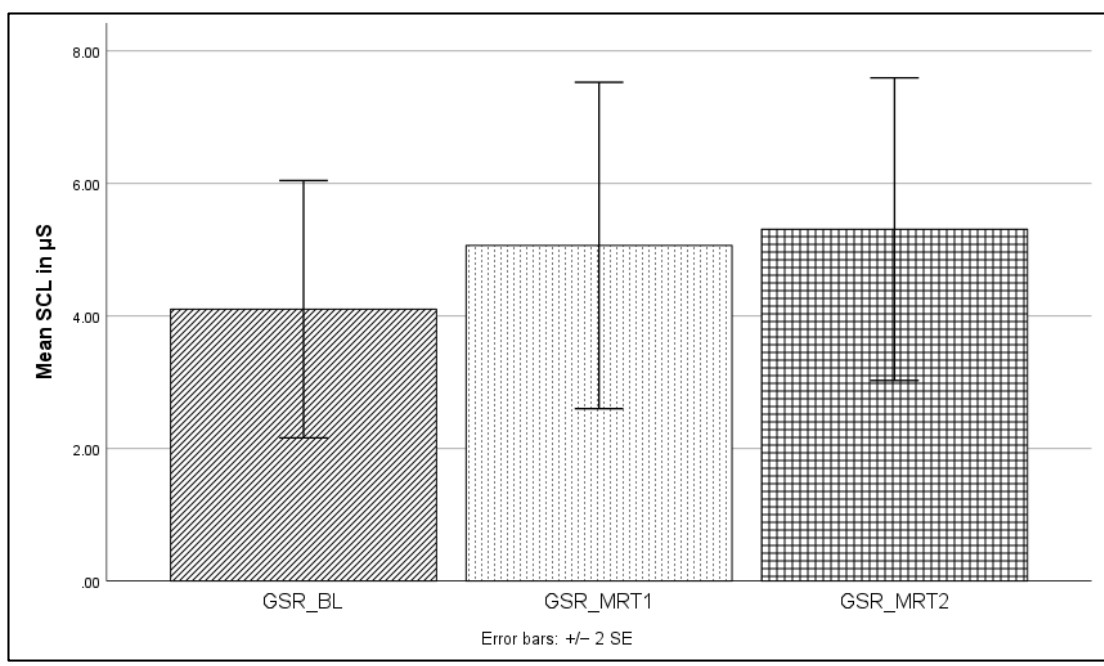

**Figure 2.** Differences in mean skin conductance level (SCL) across three test conditions: pre-test baseline (GSR_BL), first part of the test (GSR_MRT1) and second part of the test (GSR_MRT2).

### 3.2. The Relationship between Skin Conductance, Accuracy, Reaction Time and Item Difficulty (H2)

A Pearson correlation coefficient found that there was a strong positive significant relationship between SCL measured by GSR ($M = 7.23$, $SD = 7.74$) and scores on items with abstract stimuli rotated in-depth ($M = 1.23$, $SD = 3.95$) ($r(21) = 0.67$, $p < 0.001$). This relationship can account for 67% of variation in scores. Moreover, there was a strong positive correlation between participant reaction time (RT) ($M = 16.83$, $SD = 6.50$) and scores on items with abstract stimuli rotated in-depth ($M = 1.23$, $SD = 3.95$) ($r(21) = 0.512$, $p = 0.009$). This relationship can account for 51% of the variation in scores. See Table 3.

**Table 3.** Pearson coefficient correlations between scores on items with abstract stimuli rotated in-depth (Mean_Diff), reaction time (Mean_RT) and skin conductance (GSR_Test) during the MR task.

|  |  | Mean_Diff | Mean_RT | GSR_Test |
|---|---|---|---|---|
| Pearson Correlation | Mean_Diff | 1.000 | 0.510 | 0.667 |
|  | Mean_RT | 0.510 | 1.000 | 0.439 |
|  | GSR_Test | 0.667 | 0.439 | 1.000 |
| Sig. (1-tailed) | Mean_Diff |  | 0.005 | <0.001 |
|  | Mean_RT | 0.005 |  | 0.016 |
|  | GSR_Test | 0.000 | 0.016 |  |
| N | Mean_Diff | 24 | 24 | 24 |
|  | Mean_RT | 24 | 24 | 24 |
|  | GSR_Test | 24 | 24 | 24 |

Both variables were then entered into a multiple regression, which was used to test whether participant SCL measured by GSR and reaction time (RT) were predictors of scores on items containing abstract stimuli rotated in-depth. The results of the regression indicated that the two predictors explained 43% of the variance ($R^2 = 0.46$, $F(2, 21) = 10.64$, $p < 0.001$). It was found that SCL measured by GSR predicted scores on abstract objects rotated in-depth ($\beta = 0.55$, $p = 0.004$, 95% $CI = 0.10–0.47$) but no statistically significant linear dependence of the mean of scores on abstract objects rotated in-depth on RT was detected. See Table 4.

**Table 4.** Model summary: skin conductance (GSR_Test) and reaction time (Mean_RT) predict scores on items with abstract stimuli rotated in-depth (Mean_Diff).

| | *ANOVA* [a] | | | | | |
|---|---|---|---|---|---|---|
| | **Model** | **Sum of Squares** | **df** | **Mean Square** | **F** | **Sig.** |
| 1 | Regression | 187.825 | 2 | 93.912 | 10.638 | <0.001 [b] |
| | Residual | 185.396 | 21 | 8.828 | | |
| | Total | 373.221 | 23 | | | |

[a] Dependent variable: Mean_Diff. [b] Predictors: (constant), GSR_Test, Mean_RT.

### 3.3. Differences in Accuracy and Reaction Time between SCL Groups on Rotational Axis, Stimulus Type and on Gendered Objects (H3)

A one-way repeated-measures ANOVA using the Greenhouse–Geiser correction found that there was no significant differences in accuracy scores between SCL groups on concrete and abstract stimuli and stimuli rotated in picture-plane or in-depth ($F$(3, 57) = 0.952, $p$ = 0.444, $\eta^2$ = 0.04). In relation to the main effect, there was a significant difference in accuracy scores on concrete and abstract stimuli and stimuli rotated in-depth and in picture-plane ($F$ (3, 57) = 6.55, $p$ = 0.007) with a small effect size ($\eta^2$ = 0.26). Therefore, rotational axis and stimulus type explain 26% of the variance in accuracy scores on the MR task. GSR level did not significantly explain variance in accuracy scores in this context. See Table 5.

**Table 5.** Tests of within-subjects effects of stimulus type and rotational axis (Rotation_Type) across two SCL groups (GSR_Levels) on accuracy on the MR task.

| **Measure: Accuracy** | | | | | | | |
|---|---|---|---|---|---|---|---|
| **Source** | | **Type III Sum of Squares** | **df** | **Mean Square** | **F** | **Sig.** | **Partial Eta Squared** |
| Rotation_Type | Sphericity Assumed | 0.172 | 3 | 0.057 | 6.546 | <0.001 | 0.256 |
| | Greenhouse–Geisser | 0.172 | 1.579 | 0.109 | 6.546 | 0.007 | 0.256 |
| Rotation_Type * GSR_Levels | Sphericity Assumed | 0.020 | 3 | 0.007 | 0.768 | 0.517 | 0.039 |
| | Greenhouse–Geisser | 0.020 | 1.579 | 0.013 | 0.768 | 0.444 | 0.039 |
| Error (Rotation_Type) | Sphericity Assumed | 0.500 | 57 | 0.009 | | | |
| | Greenhouse–Geisser | 0.500 | 30.005 | 0.017 | | | |

* Indicates all main effects and interactions among the variables Rotational Axis and Stimulus Type (Rotation_Type) and Skin Conductance Levels (GSR_Levels).

More specifically, pairwise comparisons highlighted accuracy in stimuli rotated in-depth was significantly lower than in stimuli rotated in picture-plane ($M$ = −0.127, $p$ = 0.054, CI (95%) −0.251−−0.002). Furthermore, there was a tendency for accuracy on abstract stimuli to be higher than on stimuli rotated in-depth ($M$ = 0.067, $p$ = 0.057, CI (95%) −0.001−0.135). See Figure 3.

A one-way repeated-measures ANOVA using the Greenhouse–Geiser correction found that there was no significant differences in reaction time between SCL groups on concrete and abstract stimuli and stimuli rotated in the picture-plane or in-depth ($F$(3, 57) = 0.006, $p$ = 0.983, $\eta^2$ = 0.00). In relation to the main effect, there was a significant difference in reaction time on concrete and abstract stimuli and stimuli rotated in-depth and in picture-plane ($F$(3, 57) = 4.37, $p$ = 0.030) with a small effect size ($\eta^2$ = 0.19). Therefore, stimulus type and rotational axis explain 19% of the variance in reaction time on the MR task.

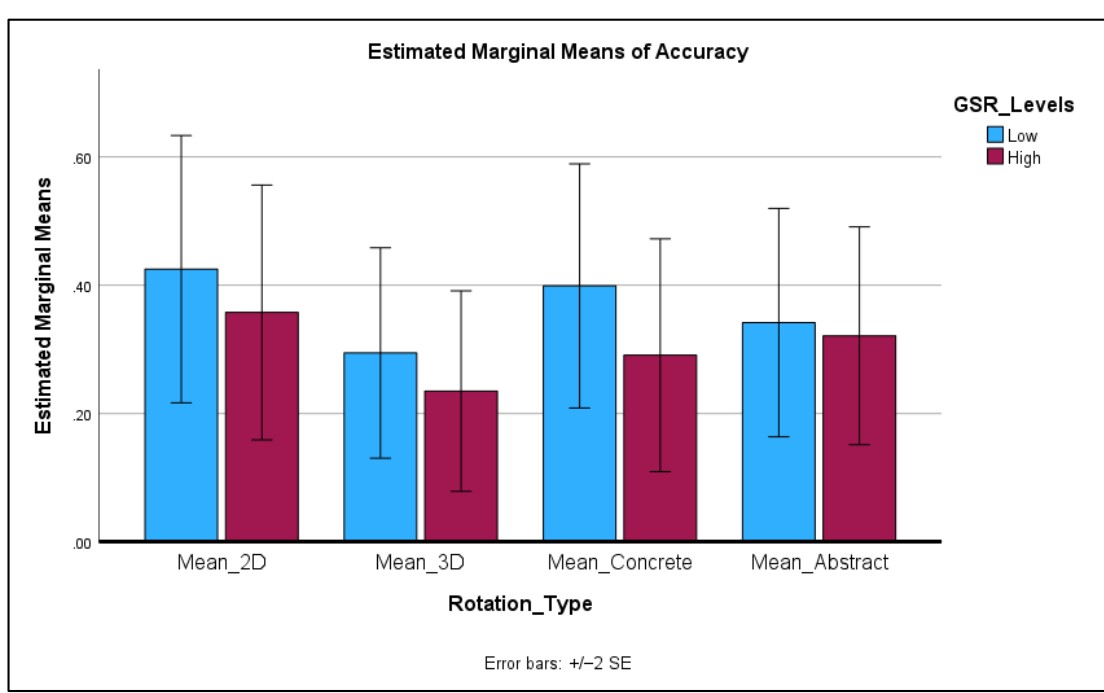

**Figure 3.** Differences in accuracy on stimuli rotated in-depth and in picture-plane and on concrete and abstract stimuli during the MR task dependent on GSR.

More specifically, pairwise comparisons highlighted that reaction time on abstract stimuli was significantly higher than on concrete stimuli ($M = -1.21$, $p = 0.041$, *CI* (95%) 0.103–0.7.22), but there was no significant difference in reaction time between stimuli rotated in picture-plane and stimuli rotated in-depth. See Figure 4.

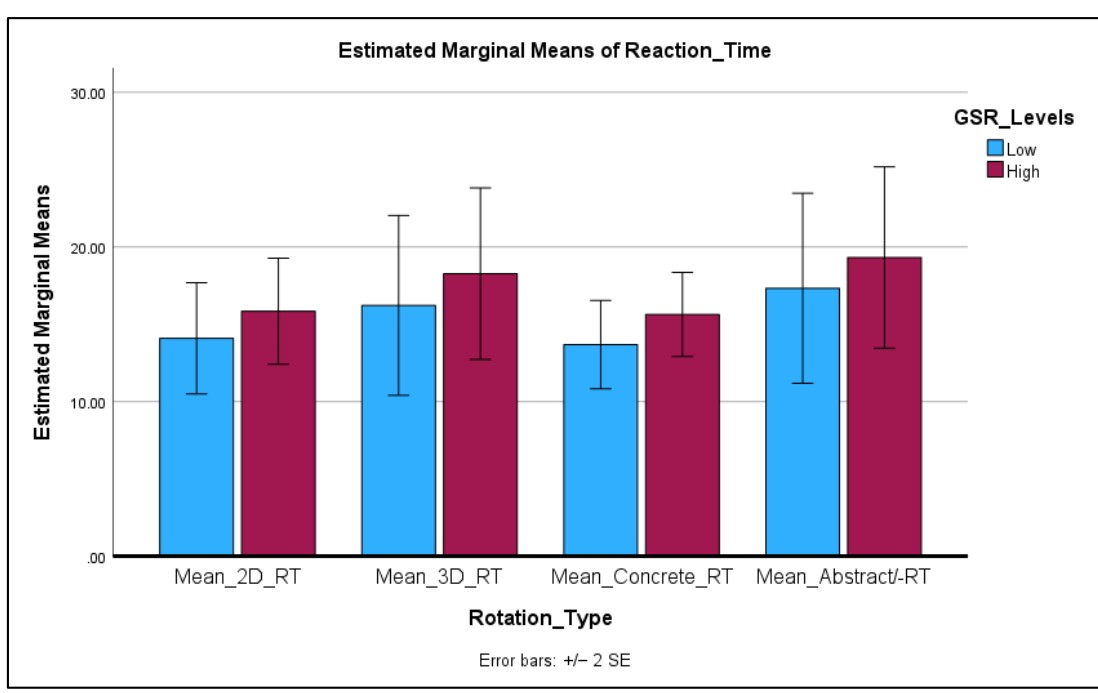

**Figure 4.** Differences in reaction time on stimuli rotated in-depth and in picture-plane and on concrete and abstract stimuli during the MR task dependent on GSR.

A one-way repeated-measures ANOVA using the Greenhouse–Geiser correction found that there was no significant differences in accuracy between SCL groups on the three

gendered objects ($F(2, 38) = 0.585$, $p = 0.553$, $\eta^2 = 0.03$). In relation to the main effect, there was no significant difference in accuracy on the three gendered objects ($F (2, 38) = 0.584$, $p = 0.553$, $\eta^2 = 0.03$).

A one-way repeated-measures ANOVA using the Greenhouse–Geiser correction found that there was a tendency toward significant differences in reaction time between SCL groups on gender-stereotyped objects ($F(2, 38) = 2.91$, $p = 0.092$, $\eta^2 = 0.13$). In relation to the main effect, there was a significant difference in reaction time on the three gendered objects ($F (2, 38) = 6.19$, $p = 0.014$) with a small effect size ($\eta^2 = 0.25$). Therefore, the three gendered objects explain 25% of the variance in reaction time on the MR task. See Table 6.

**Table 6.** Tests of within-subjects effects of gendered objects (Gendered_Obj) across two SCL groups (GSR_Levels) on reaction time on the MR task.

| Measure: Reaction_Time | | | | | | | |
|---|---|---|---|---|---|---|---|
| Source | | Type III Sum of Squares | df | Mean Square | F | Sig. | Partial Eta Squared |
| Gendered_Obj | Sphericity Assumed | 148.545 | 2 | 74.272 | 6.192 | 0.005 | 0.246 |
| | Greenhouse–Geisser | 148.545 | 1.293 | 114.848 | 6.192 | 0.014 | 0.246 |
| Gendered_Obj * GSR_Levels | Sphericity Assumed | 69.729 | 2 | 34.864 | 2.906 | 0.067 | 0.133 |
| | Greenhouse–Geisser | 69.729 | 1.293 | 53.911 | 2.906 | 0.092 | 0.133 |
| Error (Gendered_Obj) | Sphericity Assumed | 455.829 | 38 | 11.995 | | | |
| | Greenhouse–Geisser | 455.829 | 24.575 | 18.549 | | | |

\* Indicates all main effects and interactions among the variables Gendered Objects (Gendered_Obj) and Skin Conductance Levels (GSR_Levels).

Moreover, pairwise comparisons highlighted reaction time (RT) on neutral objects was significantly higher than on female-stereotyped objects ($M = 3.73$, $p = 0.004$, *CI* (95%) 1.15–6.30). However, RT on neutral objects was not significantly higher than on the male-stereotyped objects ($M = 2.33$, $p = 0.334$, *CI* (95%) −1.34–6.00) See Table 7 and Figure 5.

**Table 7.** Pairwise comparisons with interactions between skin conductance conditions (SCL levels) and reaction time (RT) on items with gender-stereotyped stimuli: male-, female- and neutral objects.

| Measure: Reaction_Time | | | | | | |
|---|---|---|---|---|---|---|
| (I) Gendered_Obj | (J) Gendered_Obj | Mean Difference (I-J) | Std. Error | Sig. [a] | 95% Confidence Interval for Difference [a] | |
| | | | | | Lower Bound | Upper Bound |
| 1 | 2 | 1.394 | 0.722 | 0.205 | −0.500 | 3.289 |
| | 3 | −2.332 | 1.397 | 0.334 | −6.000 | 1.336 |
| 2 | 1 | −1.394 | 0.722 | 0.205 | −3.289 | 0.500 |
| | 3 | −3.726 * | 0.981 | 0.004 | −6.302 | −1.151 |
| 3 | 1 | 2.332 | 1.397 | 0.334 | −1.336 | 6.000 |
| | 2 | 3.726 * | 0.981 | 0.004 | 1.151 | 6.302 |

Based on estimated marginal means. * The mean difference is significant at the 0.05 level. [a] Adjustment for multiple comparisons: Bonferroni.

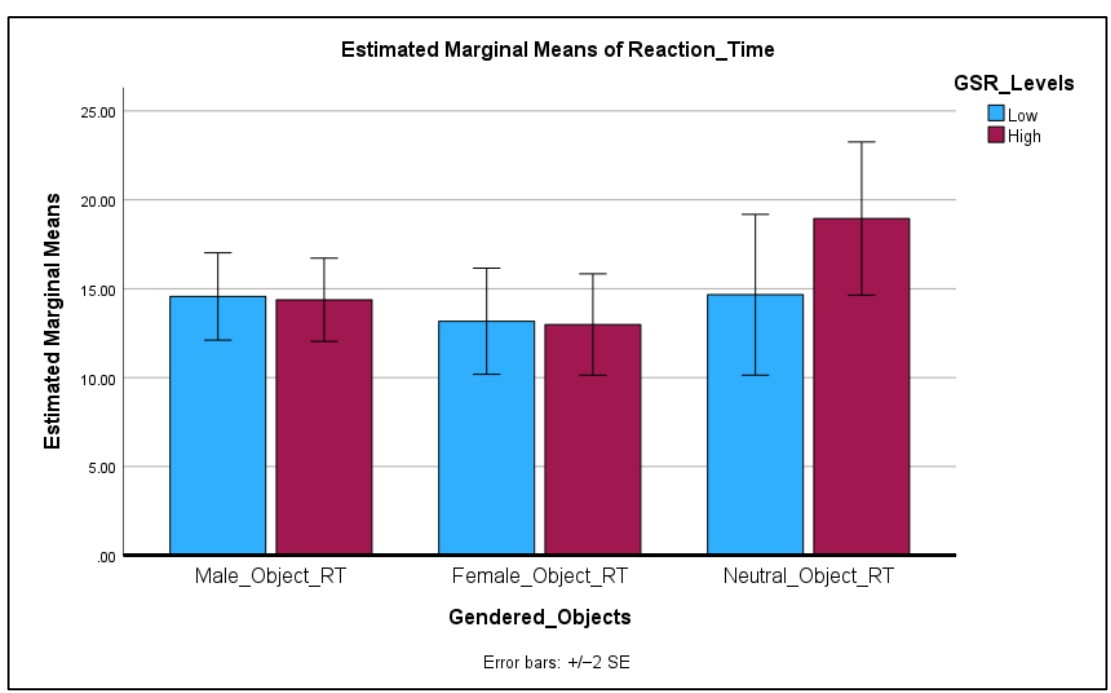

**Figure 5.** Differences in reaction time across two SCL levels (GSR_Levels) on gendered objects during the MR task.

*3.4. Gender Stereotyped Nature of Stimuli and Difficulty of Stimuli Questionnaire Differences (H4 and H5)*

Responses on the gender-stereotyped nature of stimuli questionnaire descriptively show that more students rated the *Car* ($M = 1.88$, $SD = 0.74$) and the *Cube figure* ($M = 2.52$, $SD = 1.05$) stimuli as more masculine. Other stimuli, that is, the *Letter_g*, the *Stag*, the *Pellet* and the *Polyhedron* figure presented in the questionnaire were rated as more neutral ($M = 3.05$, $SD = 0.87$).

Responses on the perceived difficulty of stimuli questionnaire descriptively show that more students rated the *Letter_F* ($M = 1.58$, $SD = 0.83$), the *Crocodile* ($M = 2.45$, $SD = 0.71$) and the *Hammer* ($M = 2.50$, $SD = 1.02$) stimuli, all of which were rotated in the picture-plane, as easier. As expected, concrete stimuli ($M = 2.17$, $SD = 0.85$) were rated easier than abstract stimuli ($M = 3.03$, $SD = 0.84$).

A two-way repeated measures ANOVA using the Greenhouse Geiser correction found that there were significant differences in accuracy ($F(4, 68) = 4.61$, $p = 0.012$, $\eta^2 = 0.21$) and reaction time ($F(4, 68) = 4.30$, $p = 0.029$, $\eta^2 = 0.20$) on the items identified as masculine in the stereotyped nature of stimuli questionnaire and those items identified as easier in the perceived difficulty questionnaire, explaining 21% of the variance in accuracy and 20% of the variance in reaction time on these items. See Table 8.

Upon inspection of the means for accuracy and reaction time on these stimuli, it was found that the *Hammer* had the highest mean accuracy ($M = 0.56$, $SD = 0.511$) and the highest reaction time ($M = 16.12$, $SD = 5.37$). The *Car* had the lowest mean accuracy ($M = 0.17$, $SD = 0.383$) and a high reaction time ($M = 13.94$, $SD = 0.383$). The *Crocodile* had a high mean accuracy ($M = 0.50$, $SD = 0.514$) and the lowest reaction time ($M = 9.74$, $SD = 4.01$). The *Letter_F* had mean accuracy ($M = 0.44$, $SD = 0.511$) and mean reaction time ($M = 11.83$, $SD = 4.17$). The *Cube figure* had mean accuracy ($M = 0.39$, $SD = 0.502$) and mean reaction time ($M = 14.91$, $SD = 7.35$). See Figure 6.

**Table 8.** Differences in accuracy and reaction time (RT) on items identified in the stereotyped nature of stimuli and perceived difficulty questionnaire.

| Source | | Measure | | Type III Sum of Squares | df | Mean Square | F | Sig. | Partial Eta Squared |
|---|---|---|---|---|---|---|---|---|---|
| factor1 | Accuracy | | Sphericity Assumed | 1.622 | 4 | 0.406 | 4.613 | 0.002 | 0.213 |
| | | | Greenhouse–Geisser | 1.622 | 2.372 | 0.684 | 4.613 | 0.012 | 0.213 |
| | RT | | Sphericity Assumed | 472.286 | 4 | 118.072 | 4.301 | 0.004 | 0.202 |
| | | | Greenhouse–Geisser | 472.286 | 2.678 | 176.377 | 4.301 | 0.012 | 0.202 |
| error (factor1) | Accuracy | | Sphericity Assumed | 5.978 | 68 | 0.088 | | | |
| | | | Greenhouse–Geisser | 5.978 | 40.318 | 0.148 | | | |
| | RT | | Sphericity Assumed | 1866.817 | 68 | 27.453 | | | |
| | | | Greenhouse–Geisser | 1866.817 | 45.521 | 41.010 | | | |

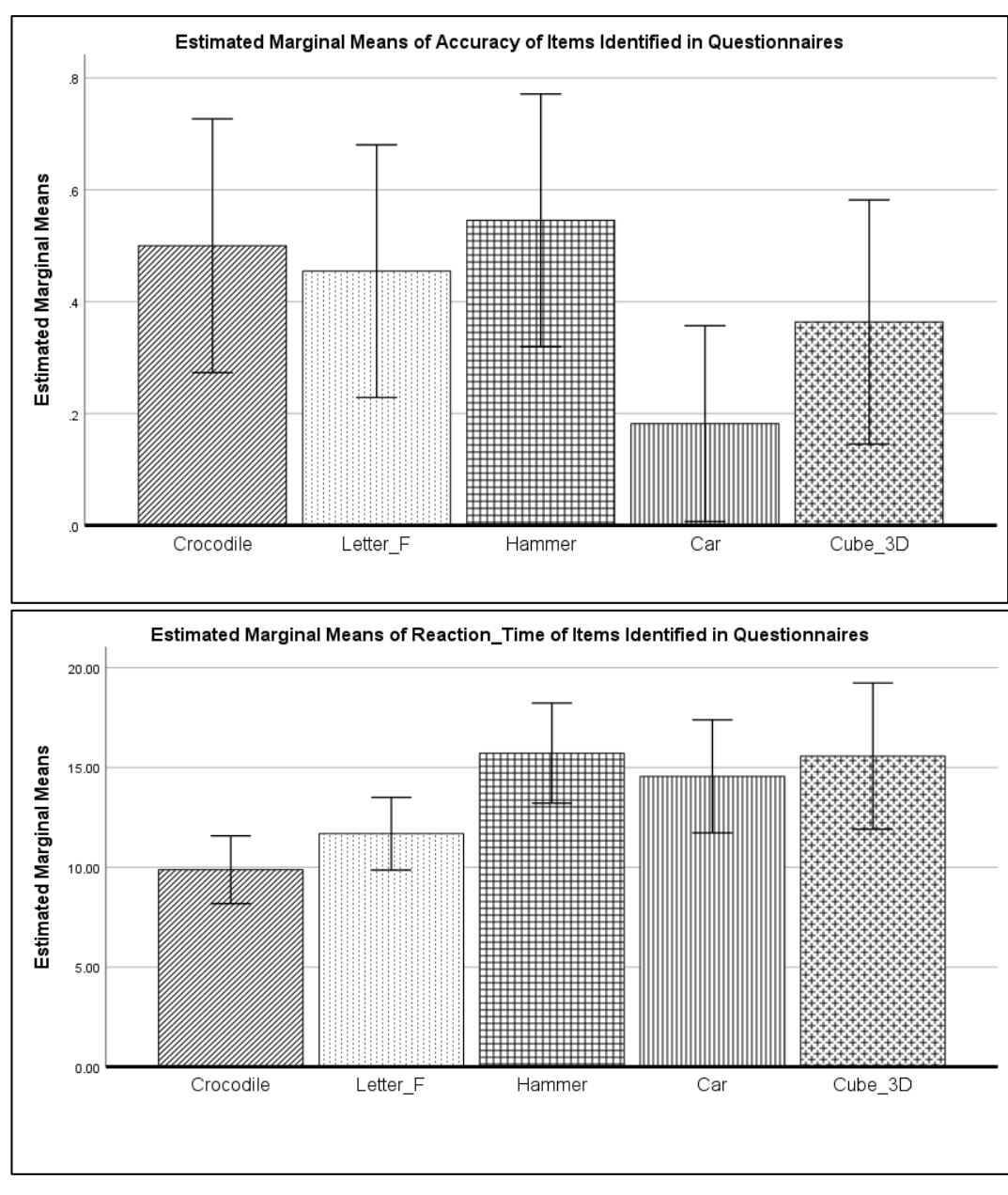

**Figure 6.** Differences in accuracy and reaction time on objects identified in the stereotyped nature of stimuli and perceived difficulty questionnaires.

## 4. Discussion

This pilot study set out to investigate whether emotional regulation measured by skin conductance has an impact on primary school children's performance on an MR task with a view to using the results to support the development of a computer adaptive test of mental rotation.

### 4.1. Skin Conductance Levels across the MR Task

Firstly, results demonstrate that skin conductance levels (SCL) did indeed change as soon as the MR task began. There was a significant increase in SCL from baseline measurements to part one of the task (MRT1), which was considered the easier part as stimuli were rotated in picture-plane only. Moreover, SCL on both parts of the task was significantly higher than in the baseline condition. Contrary to expectations, there was, however, no significant difference in SCL between the first part of the task (MRT 1) and the second, which was considered the more difficult part due to stimuli being rotated in-depth (MRT 2). Therefore, in this sample, an increased task difficulty had no significant effect on emotional reactivity measured by SCL in the second part of the MR task. There were, however, clear groups of participants who experienced "high" and "low" SCL levels during the MR task and these states influenced their performance on some items. These results will now be discussed.

### 4.2. The Relationship between Skin Conductance, Accuracy, Reaction Time and Item Difficulty

It is known from previous research that items containing abstract stimuli rotated in-depth are more difficult to solve on an MR task than items with concrete stimuli rotated in picture-plane. Therefore, this study also investigated the relationship between item difficulty, emotional regulation measured by SCL and reaction time (RT) on the MR task. The results demonstrate that there was a positive relationship between emotional regulation measured by SCL, RT and scores on difficult items (considered all items across the full MRT containing abstract stimuli rotated in-depth) in this study. Indeed, as emotional regulation measured by SCL and RT increased, so did scores on these items. Therefore, it appears that performance on the difficult items benefitted from increases in participant effort and time spent on these items. Furthermore, findings of this study also show that emotional regulation measured by SCL, in particular along with RT, were significant predictors of scores on difficult items. Thus, in this sample, participants who were able to regulate emotions effectively and who spent more time trying to determine the correct solution to more difficult items benefitted in that they achieved better outcomes.

### 4.3. Differences in Accuracy and Reaction Time between SCL Groups on Rotational Axis, Stimulus Type and on Gendered Objects

Individual SCL varied, and likewise in this sample. There were participants with a higher SCL than others. Therefore, the study endeavored to examine if the rotational axis (whether rotated in picture-plane or in-depth), the type of stimuli (whether concrete or abstract) and the gender-stereotyped nature of the stimuli (whether male-stereotyped, female-stereotyped or gender-neutral objects) would lead to differences in accuracy and reaction time across groups of "high" and "low" SCL participants. It was found that rotational axis, stimulus type and the gender-stereotyped nature of stimuli did lead to significant differences in accuracy as a main effect, but they did not significantly interact with SCL. Therefore, as SCL differs individually, each participant's emotional reactivity measured by SCL appears to have adjusted accordingly during the cognitive task so that it did not result in differences in performance. Therefore, being a "high" or "low" SCL individual did not impact accuracy and did not interact with performance by stimulus type, rotational axis or gender-stereotyped stimuli.

It was, however, found that there was a tendency toward significant differences in reaction time (RT) on gender-stereotyped objects. Moreover, this difference was significant

between SCL groups. Specifically, RT in the "high" SCL group was significantly higher on the gender-neutral objects than on the female-stereotyped objects.

There are two possible explanations for this result. Emotional arousal increases when an individual is presented with stimulus which is relevant and salient to them. Through this increase, the observer assigns a higher priority to that stimulus. It may therefore be that objects represented in the gender-neutral stimuli were more relevant to participants in this study. These stimuli contained everyday objects, which also have recreational or educational significance, such as the scissors, the bicycle and the pot. Participants were likely very familiar with these objects, hence, they were relevant and important to them, resulting in more time spent at attempting to solve the MR test items containing these objects. The female-stereotyped objects on the other hand, such as the bow, the pram and the handbag, may have been assigned a lower relevance and priority by participants because they do not regularly use these objects or they are devalued due to negative associations arising from stereotype threat. This may have resulted in a tendency for participants to dismiss the items containing these objects more quickly.

Another explanation for longer RT on the gender-neutral objects, but one which was not investigated in this study, is the role of occlusion. Two-dimensional representations of three-dimensional objects do not allow the viewer to scan the object from multiple angles. This results in parts of it being hidden. Hidden or occluded parts of objects represented using two-dimensional media such as paper-and-pencil tests or tests undertaken on a computer screen or mobile device make it difficult for the participant to know the shape of the object without seeing it from another angle. The resulting occlusion is not only known to contribute to item difficulty and longer reaction times, but also to gender differences in performance on MR tasks (Felix et al. 2011; Nolte et al. 2022).

### 4.4. Gender-Stereotyped Objects and Perceived Difficulty of Stimuli

As expected, participants identified the 'Car' and the 'Cube', which are often classified as male-stereotyped and associated with construction and transportation activities (often referred to as "boys' toys") as more masculine (Ruthsatz et al. 2019). On the perceived difficulty questionnaire, the 'Hammer', the 'Crocodile' and the 'Letter_F' were rated as easier. Items with animals and letters are found on the picture-rotation test, which was designed to test mental rotation skills in pre-school children (Quaiser-Pohl 2003).

Upon an analysis of accuracy and reaction time during the MR task on objects identified in the stereotyped nature of stimuli and the perceived difficulty questionnaires, there were significant differences found within the sample. Participants achieved higher scores and spent more time on the male-stereotyped 'Hammer'. Similarly, the highest reaction time was found on the male-stereotyped 'Car'. These objects appear to have been highly salient and participants may therefore have prioritized them by spending more time attempting to solve them. As expected, easy items such as the 'Crocodile' were solved accurately and quickly in this sample.

### 4.5. Relevance for the Development of a CAT-MR

In this study, items with abstract and gendered stimuli and objects rotated in-depth appeared to be more difficult and required a longer amount of time to solve than others types of items, a result also found in previous research (Neuburger et al. 2011, 2012b; Ruthsatz et al. 2017).

Salience and relevance of stimuli are also important factors when choosing items containing these objects for a CAT, as they may be given higher priority by participants, resulting in longer reaction times and, hence, in many cases, a higher accuracy. Similarly, items with stimuli considered easier or less relevant to participants may result in shorter reaction times, which may then impact accuracy or the ability to correctly solve items containing these objects.

Although a purely anecdotal observation in this study, the visibility of features of some stimuli, such as the hidden edges of cube figures or gender-neutral objects, has been found

to be related to the level of difficulty in solving them. Therefore, presenting the stimuli in another format such as real-life objects or three-dimensional models on the computer screen may improve accuracy. Such items could also be incorporated into a CAT.

Item banks in CAT are files of various suitable test items that are coded by subject area, instructional level and other pertinent item characteristics such as item difficulty and discriminating power (Gronlund 1998). In creating an item bank for the CAT-MR, the results of this and any consequent replicated study regarding those items to be used in a CAT-MR need to be considered.

*4.6. Limitations and Outlook*

This was a pilot study, so the sample size was a significant limitation. Therefore, the study is now being replicated with a bigger sample size.

As occlusion may be limiting participants' performance on the MR task, it may be useful to develop an MR task using real-life objects or three-dimensional models of stimuli to be used in future experiments.

No data regarding test or spatial anxiety were collected, but this could be beneficial in providing more insight into participants' experience of and their feelings about the task. Furthermore, academic self-concept could provide useful data on baseline perceived academic ability in primary school children. Self-concept questions regarding mathematical ability are included in this questionnaire and could yield valuable additional information. Mathematical ability is highly correlated with spatial ability in adults and children (Rahe and Quaiser-Pohl 2021)

There was unfortunately a loss of data due to technical issues. This has now been resolved so that it should not occur during the larger data collection. An additional physiological method of measuring emotional arousal during an MR task has been added for the main study.

Due to the above-mentioned limitations, a number of queries arose, which the pilot study was unable to answer, but which are important for the development of the CAT-MR: (a) whether the MR task was too difficult and too long for some participants; (b) the issue of how the test was constructed, i.e., gender-stereotyped items and increasing item difficulty in the second part of the task; (c) whether some participants may have performed better on a test in which item selection is adapted to their ability, e.g., a CAT-MR; and (d) whether taking a CAT-MR might impact positively on emotional reactivity, accuracy and speed. These queries will be addressed in more detail in the larger study.

## 5. Conclusions

Reducing the disadvantages in cognitive testing is paramount to preserving the reliability and validity of psychometric instruments designed to measure spatial abilities such as mental rotation. Moreover, providing individuals exposed to stereotype threat with a means to cope effectively with negative emotions such as anxiety can restore executive resources, improve cognition and thus, test performance (Johns et al. 2008). This can also lead to diminishing the gender difference in spatial test performance (Voyer et al. 1995; Wraga et al. 2006). Therefore, the identification of the effects of gender-stereotyped stimuli and emotional arousal arising during testing, as well as item difficulty and the chronological order of items on spatial ability performance warrant ongoing investigation. This research is key to ensuring that these factors do not compromise measurement accuracy nor contribute to increasing gender differences, but rather serve to measure spatial ability accurately, regardless of gender. Test construction as a science benefits from such research as does the field of gender and STEM. In the pursuit of the development of psychological assessment approaches such as a CAT-MR, the authors seek to identify and decrease bias in tests on spatial ability, thus contributing positively to the reduction in the gender gap in STEM.

**Supplementary Materials:** PsychoPy® is a free, cross-platform, open-source package allowing researchers to run a wide range of experiments in the behavioral sciences. It can be downloaded at www.psychopy.org/download.html (accessed on 6 April 2023).

**Author Contributions:** Conceptualization, M.L.-M. and C.M.Q.-P.; methodology, M.L.-M.; software, M.S.; validation, M.L.-M. and C.M.Q.-P.; formal analysis, M.L.-M.; investigation, M.L.-M.; resources, M.S. and V.R.; data curation, M.L.-M.; writing—original draft preparation, M.L.-M.; writing—review and editing, M.L.-M. and C.M.Q.-P.; visualization, M.L.-M.; supervision, C.M.Q.-P.; project administration, M.L.-M.; funding acquisition, C.M.Q.-P. All authors have read and agreed to the published version of the manuscript.

**Funding:** This research was funded by the Marie Sklodowska-Curie Actions International Training Network (MSCA ITN) "SellSTEM", grant number 956124.

**Institutional Review Board Statement:** The study was conducted in accordance with the Declaration of Helsinki, and approved by the Institutional Review Board (or Ethics Committee) of the University of Koblenz, Germany (Ethikkommission/Antrag Lennon-Maslin/Beschluss vom 9 November 2022). Furthermore, according to section 67, paragraph 6 of the Education Act of the State of Rhineland-Palatinate, Germany, approval was sought and provided by the Aufsichts- und Dienstleistungsdirektion Rheinland Pfalz (ADD RLP), Willy-Brandt-Platz 3, 54290 Trier, Germany, the authorities who oversee schools in this state.

**Informed Consent Statement:** Informed consent was obtained from all participants involved in the study.

**Data Availability Statement:** The data presented in this study are available on request from the corresponding author. The data are not publicly available due to ethical and privacy reasons stated in the parental consent information form.

**Acknowledgments:** Many thanks to Martina Rahe for support and advice regarding the statistical analyses. A special thanks to Emil Ruthsatz, who kindly tested a trial version of our experiment before we undertook the study in a school. Thanks also to Heike Ruf-Urmersbach, who provided advice and support regarding the schools, and to our research assistant Annika Volkmann, who has been a consistent support for our project throughout.

**Conflicts of Interest:** The authors declare no conflict of interest.

## Appendix A. Stereotype Questionnaire

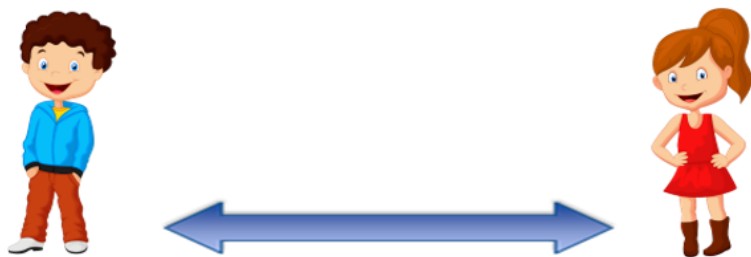

On a sliding 5-point scale, participants were asked to rate how gender-stereotyped they considered the stimuli used in the MR task.

## Appendix B. Difficulty Questionnaire

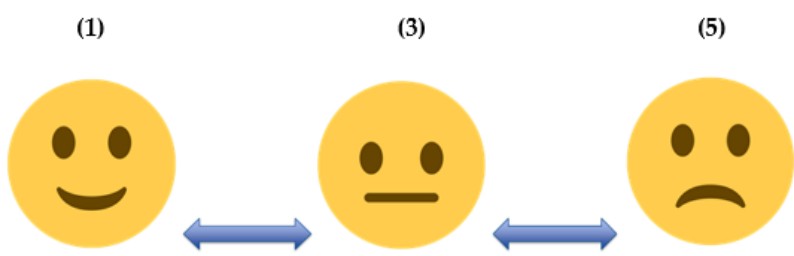

On a sliding 5-point scale, participants were asked to rate how difficult they found various items on the MR task.

**Appendix C. Items Used in the MR Task**

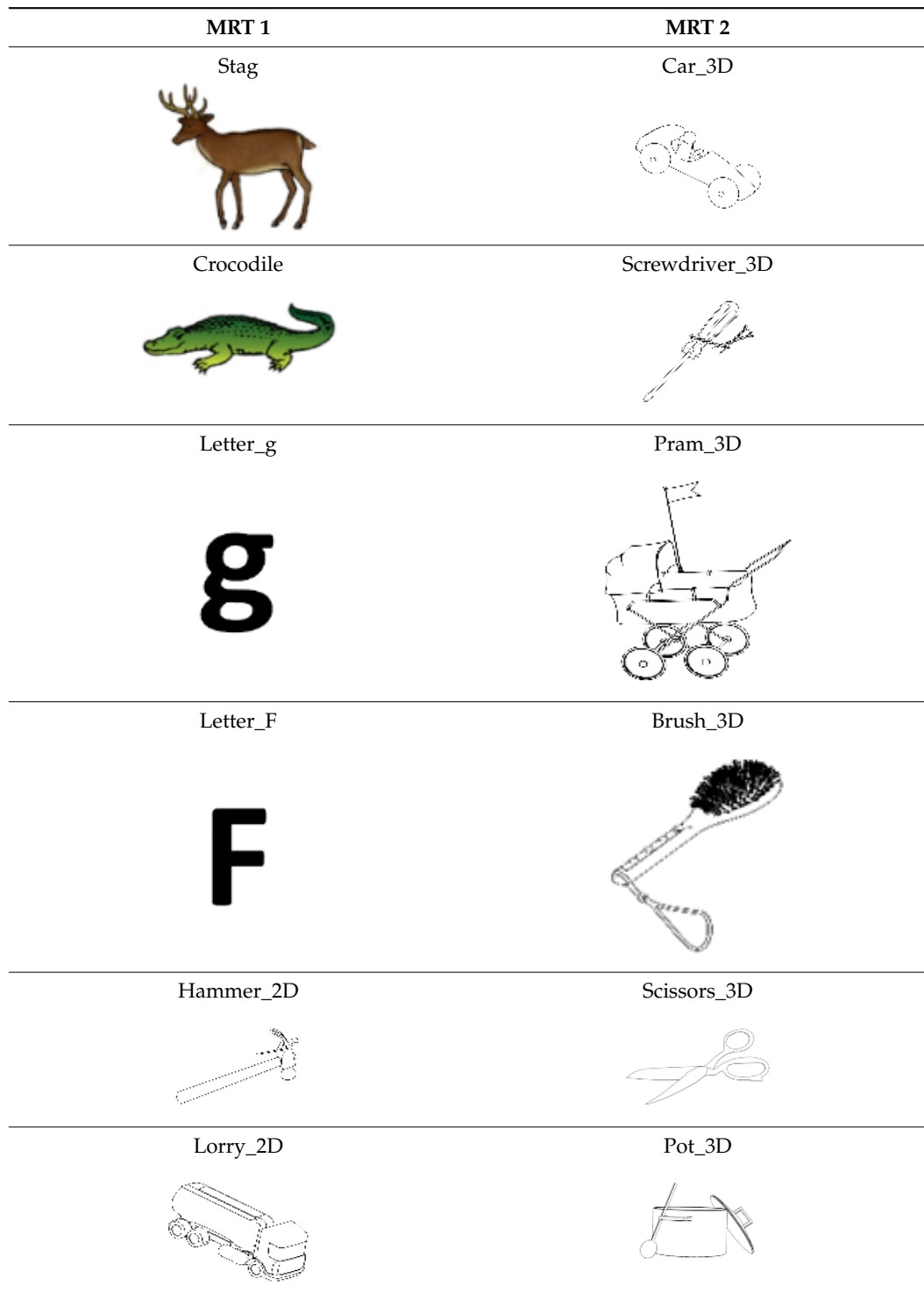

| MRT 1 | MRT 2 |
| --- | --- |
| Stag | Car_3D |
| Crocodile | Screwdriver_3D |
| Letter_g | Pram_3D |
| Letter_F | Brush_3D |
| Hammer_2D | Scissors_3D |
| Lorry_2D | Pot_3D |

| MRT 1 | MRT 2 |
|-------|-------|
| Handbag_2D | Cubes_3D_#5 |
| Bow_2D | Cubes_3D_#3 |
| Bike_2D | Pellets_3D_#1 |
| Goggles_2D | Pellets_3D_#5 |
| Cubes_2D_#1 | Polyhedron_3D_#1 |
| Cubes_2D_#4 | Polyhedron_3D_#11 |
| Pellets_2D_#1 | |
| Pellets_2D_#5 | |

| MRT 1 | MRT 2 |
|---|---|
| Polyhedron_2D_#11 | |

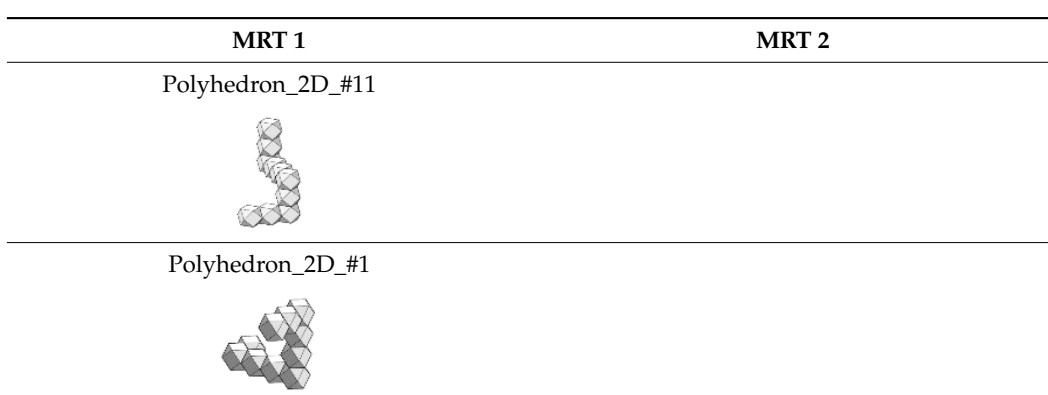

| | |
|---|---|
| Polyhedron_2D_#1 | |

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
