# Peer review of "Under My Skin: Reducing Bias in STEM through New Approaches to Assessment of Spatial Abilities Considering the Role of Emotional Regulation"

_socsci, doi:10.3390/socsci12060356_

Round 1

Reviewer 1 Report (New Reviewer)

Data collection procedures were thoroughly described and illustrated. Outstanding documentation of repeated-measures ANOVA results, with related visuals. I also appreciated the specificity of recommendations for future related research to mitigate identified limitations in future replications of the study. Very well written and worthy of publication.

Author Response

Dear Reviewer,

Thank you for the time taken to assess my manuscript and for your positive evaluation. I really appreciate it. It is very encouraging in my pursuit of the larger study.

Many thanks again.

Kind regards

Author 1

Reviewer 2 Report (New Reviewer)

Review of “Under my Skin: Reducing Bias in STEM through New Approaches to Assessment of Spatial Abilities Considering the Role of Emotional Regulation”

27 Apr 2023

This article describes the first part of a study series; this part focuses on determining if emotional regulation in ~9-year-olds affects performance on mental rotation tasks.

I find the idea of the study interesting and useful. The whole topic of mental rotation has been used and mis-used over the last few decades to explain boys’ and men’s higher interest in STEM fields. This study suggests that emotional arousal and regulation can affect performance and may explain large portions of the gender gap shown in (some) mental rotation tasks.

I think that the study design and analysis is reasonable. My stats is far enough in the past that unfortunately I can’t speak to the validity of the choices made and data presented.

My biggest issue with the manuscript is the intro and lit review. Having looked up many of the references studies, I find that many of them are not represented properly or are reframed in a different way than intended.

The intro and lit review need to be significantly revised to match the results suggested in the text. As examples:

--Line 40: Only one field is mentioned (biomedical engineering).

--Line 43: Bartlett and Camba do not see significant differences favoring males.

--Line 81: Ramirez et al. is only on young children.

--Line 137: Bethell-Fox & Shepard does not match the asserted finding.

Some other pieces that would help the reader:

--Lines 306-312: what is µS? please provide the actual variable/unit.

--Abstract should include that study was on ~9 year-olds.

--Figures: what do the error bars represent? In Figure 2 the error bars suggest non-significance between the three conditions.

--Line 473: Neutral/high is higher than female, but it also looks significantly higher than male. Explain?

--I was confused at line 530: doesn’t this contradict results in section 4.1?

--Line 592: what does “prioritized” mean when problems were posed linearly?

I feel this needs significant revision before it is ready for publication. The study is interesting, but the manuscript needs work.

Minor editing needed.

Author Response

Dear Reviewer,

Many thanks for the time taken to assess my manuscript and for spotting those issues. I really appreciate it. I have now amended these as follows:

  1. "My biggest issue with the manuscript is the intro and lit review. Having looked up many of the references studies, I find that many of them are not represented properly or are reframed in a different way than intended" - I have checked and removed incorrect references or amended the text accordingly. See below:

    The intro and lit review need to be significantly revised to match the results suggested in the text. As examples:
    --Line 40: Only one field is mentioned (biomedical engineering). Clarified now in the text.
    --Line 43: Bartlett and Camba do not see significant differences favoring males. Reference removed
    --Line 81: Ramirez et al. is only on young children. Clarified in the text.
    --Line 137: Bethell-Fox & Shepard does not match the asserted finding. Reference removed.

    Some other pieces that would help the reader:
    --Lines 306-312: what is µS? please provide the actual variable/unit.- I added a footnote with an explanation of the unit microsiemens (uS) with a reference to where I found this information

    --Abstract should include that study was on ~9 year-olds - I added the mean age of participants (M = 9.28 years old) to the abstract.

    --Figures: what do the error bars represent? In Figure 2 the error bars suggest non-significance between the three conditions - I redid the chart with error bars representing the SE and added a piece interpreting what they mean.

    --Line 473: Neutral/high is higher than female, but it also looks significantly higher than male. Explain? - I added an explanation here and another table with the pairwise comparisons with significant and non-significant results.

    --I was confused at line 530: doesn’t this contradict results in section 4.1? I rewrote this part so it makes more sense and to clarify the difference between "task" difficulty and "item" difficulty.

    --Line 592: what does “prioritized” mean when problems were posed linearly? I elaborated here so it now makes more sense I believe.

Many thanks again.

Best regards

Author 1

Reviewer 3 Report (New Reviewer)

In depth analysis of the content (all major points discussed).
Excellent summary.
Communicates the key ideas/themes/findings with a high degree of clarity and insight.
 Engaging introduction and conclusion, both indicate the overall focus of the paper. Logical development of ideas through well-developed paragraphs, good use of transitions. Address minor punctuation conventions.

Address minor punctuation conventions.

Author Response

Dear Reviewer,

Thank you very much for the time taken to assess my manuscript and for your positive evaluation. I have now addressed those punctuation issues and typos. Thank you for pointing these out. I appreciate your supporting comments.

Many thanks again.

Kind regards

Author 1

This manuscript is a resubmission of an earlier submission. The following is a list of the peer review reports and author responses from that submission.

Round 1

Reviewer 1 Report

The title as well as the introduction raised expectations about your manuscript and research. The topic you are addressing would be a relevant addition to existing literature. Thank you for this valuable contribution. I will structure my feedback in (a) general remarks (these comments cover feedback applicable in the entire manuscript), and (b) specific remarks (feedback on sentence and/or word level). The specific remarks can include a quote from your original manuscript to refer to a specific section. The specific remarks will refer to page (emphasis added in boldface; e.g., 1.15/16) and row(s; e.g., 11.15/16).

General remarks:

The overall manuscript is neat and written concisely—with relevant information for existing literature. One aspect that you can focus on is the readability of your manuscript. Furthermore, the small sample size requires a careful approach. I am wondering you did not make any changes to make it a mixed method study. It would be suitable and you gain more insight in what is going on.

Specific remarks:

p.abstract        Avoid references in the abstract.

p.1.28–29        I would delete the quote; it does not add anything new to your message (you make your point).

p.1.30–36        Add funding information at the end of the manuscript. It is not crucial for you research questions.

p.2.46              “across Europe” is vague. Can you be more specific?

p.2.53/54         I do not know what kind of reference format you are using, but I am unfamiliar with listing two authors and then “et al.”. In row 56 you only list one author with “et al.”.

p.2.80–82        In one sentence you refer to your target group as children and in the last part as test takers. Keep this consistent.

p.3.137            It is a pilot study? This has to be introduced earlier in your manuscript. Moreover, I would rephrase the assumptions. The current phrasing is not suitable for assumptions/hypotheses. Easier items refer to their difficulty (you refer to this later (with v.). Furthermore, the second part of the task is vague (refer to advancing during the task, or completing the task).

p.3.158–161    This small sample allows for more qualitative research. Why wasn’t this explored (in addition to your quantitative study)? With an n = 3 (n needs to be placed in italics) your results can hardly be interpreted.

p.3.164 – 196 This is difficult to read due to the indent and short sentences. Can you display this differently (e.g., a table)? Also the quotation marks do not work in the last paragraph on that page. Moreover, depending of your reference format, you need to place the anchors in italics (rather than using quotation marks).

p.4                   Avoid bullet points.

p.5.277            The space after the “1” should be removed.

p.5.278            Avoid questions as headers.

p.5.280            The “m” needs to be place in italics. Apply this in the remainder of your work as well.

p.5                   Again the bullet points results is having less overview of your results. Structure it differently.

p.6                   I would use patterns or prints in the bars (alongside colours). Not everyone will print your manuscript and, if they do, the different patterns clearly indicate the three different bars. Also apply this to the remaining graphs.

p.7.364            “with at least one participants” à You have three participants. Be specific.

p.8.413            Again, do not pose more questions in your work. Rephrase them as statements and suggestions for future research. You can merge this paragraph with the following (limitations and outlook).

p.8.458            So the funding was blinded for review, yet the information in the manuscript was not blinded?

p.references    If you are using APA 7, you need to go over your manuscript. Make the capital letter use in the document titles consistent.

Reviewer 2 Report

The authors promise far more than they can achieve due to the limitations of their research sample. They state very clearly that it is impossible to draw any conclusion from their 3 subjects' sample study. Whereas presenting an ambitious abstract and all figures and the discussion of the study, suggest something else. I find it irresponsible. The problems of the article are important and valid for gender problems in science (STEM) issues. Additionally, the article nicely shows the present state of the art in the discussion of the methodology of mental rotation experiments. I recommend restructuring and recomposing the article. It would be far more beneficial to make the discussion around methodological issues the main focus and the study should be treated only as an illustration and the departure point for future studies.

Reviewer 3 Report

I agreed to review the paper as I had hoped that it would make a contribution to the field. However, I am a little surprised by what I read. I will list my concerns with the paper below:

1) I have been involved for decades in STEM education research and I do agree that the development of spatial reasoning and abilities is important. I had hoped this would be discussed. However, the paper focused on the testing. So instead of focusing how one can help girls develop spatial reasoning, the focus was on the bias in testing. Yes, it has been well documented that the girls lag in spatial reasoning. So I do not see how this research contributes to solving the problem.

2) The literature review and the claims about the lack of girls in STEM are rather limited and not focused on spatial reasoning. For example, in life sciences, there are more women today graduating from university (at least in North America), which is well documented. Spatial reasoning is very important in this field, and yet the girls are successful in the life science, medicine, etc. The girls lag in physical sciences and computer sciences. So to make the big statement that the girls are behind in STEM might be not entirely accurate.

3) The number of participants was 3 (2 girls and a boy), so I do not see how these very limited data would allow for any meaningful data analysis or even generalizations. To me it looked like a science fair data collection.

4) I was very surprised that with such a limited number of participants, no qualitative data were collected. The limited number of participants usually makes the researchers to revert to the mixed study design combining qualitative and quantitative research. Yet here, the researchers stopped at analyzing three participants and drawing conclusions.

5) As a scientist, I was surprised to see that no error analysis was conducted on the data. To report a mean time of the response and not to give any error bar associated with it does not look scientifically justified for me. See Table 2. How many times were the experiments repeated? What is the mean standard error? Otherwise, Table 2 doesn't make much sense to me. So if there were three participants and the did all the tasks once, it is hard to draw any conclusions from that... Not much statistically reliable data were collected.

6) The study aimed at finding evidence to support assumptions? I am trying to understand - is the word assumption here synonymous with the word hypothesis? I am not sure I understand what kind of study is it? I realize, there is an access to the tool  measuring skin conductance response, but I do not see how it relates to improving gender gap in STEM. Moreover, what was done, to me does not describe "the impact of emotional regulation on MR in the classroom" as was claimed by the authors.

7) Finally, I do not think this study has a potential, but it has to be designed more clearly with well-defined hypotheses. The authors also have to collect data on more than 3 students with a one-time measurement. 

Therefore I do not recommend this study for publication in its current form.